# A family of unconventional deubiquitinases with modular chain specificity determinants

Thomas Hermanns [1], Christian Pichlo [2], Ilka Woiwode [1], Karsten Klopffleisch [1], Katharina F. Witting[3], Huib Ovaa[3], Ulrich Baumann[2] & Kay Hofmann [1]

Deubiquitinating enzymes (DUBs) regulate ubiquitin signaling by trimming ubiquitin chains or removing ubiquitin from modified substrates. Similar activities exist for ubiquitin-related modifiers, although the enzymes involved are usually not related. Here, we report human ZUFSP (also known as ZUP1 and C6orf113) and fission yeast Mug105 as founding members of a DUB family different from the six known DUB classes. The crystal structure of human ZUFSP in covalent complex with propargylated ubiquitin shows that the DUB family shares a fold with UFM1- and Atg8-specific proteases, but uses a different active site more similar to canonical DUB enzymes. ZUFSP family members differ widely in linkage specificity through differential use of modular ubiquitin-binding domains (UBDs). While the minimalistic Mug105 prefers K48 chains, ZUFSP uses multiple UBDs for its K63-specific endo-DUB activity. K63 specificity, localization, and protein interaction network suggest a role for ZUFSP in DNA damage response.

[1] Institute for Genetics, University of Cologne, Zülpicher Str. 47a, 50674 Cologne, Germany. [2] Institute of Biochemistry, University of Cologne, Zülpicher Str. 47, 50674 Cologne, Germany. [3] Department of Chemical Immunology, Leiden University Medical Center, Einthovenweg 20, 2333 ZC Leiden, The Netherlands. Correspondence and requests for materials should be addressed to K.H. (email: kay.hofmann@uni-koeln.de)

The covalent attachment of ubiquitin to proteins via the ε-amino group of substrate lysine residues is—besides phosphorylation—the most important posttranslational modification for regulating protein signaling and homeostasis[1]. The ability of ubiquitin to target other ubiquitin molecules at various lysine residues, giving rise to ubiquitin chains of different linkage types, contributes substantially to the versatility of the ubiquitination system[2]. Deubiquitinating enzymes (DUBs) are isopeptidases that can deconjugate single ubiquitin units or entire ubiquitin chains from proteins, thereby erasing or modulating the ubiquitin signal. In addition, several DUBs are also able to cleave peptide bonds at the C terminus of ubiquitin, an activity required for processing the primary translation products of the ubiquitin genes and thus essential for the entire ubiquitination cascade. The human genome encodes approximately 100 DUB enzymes belonging to 6 different families, which exhibit distinct but overlapping cleavage preferences[3,4]. Five of these families are cysteine proteases, some of which show weak but significant sequence similarity to each other[5]. The sixth family belongs to the metalloproteases and appears to be evolutionary more ancient, since it contains bacterial and archaeal members that are likely to act on prokaryotic ubiquitin-fold proteins[6]. Besides ubiquitin, there are several ubiquitin-like modifiers (UBLs), which also require proteases for processing their immature precursors and/or for their deconjugation. Despite their mechanistic similarities, UBL proteases typically belong to families distinct from DUBs[7]: SUMO and NEDD8 are cleaved by members of the SENP/ULP family, UFM1 is cleaved by the UFSP family, while the autophagy modifiers ATG8 and ATG12 are cleaved by members of the autophagin (ATG4) family[8,9]. Besides their catalytic domains, many DUBs and UBL proteases harbor domains or motifs for recognizing the modifier to be cleaved, or the substrate from which the modifier is removed. This trend is particularly pronounced for DUBs, where the presence of multiple ubiquitin-binding domains (UBDs) can confer specificity for ubiquitin chains of a particular linkage type[10].

Here, we describe the biochemical and structural characterization of a seventh deubiquitinase family, which is distantly related to proteases for ubiquitin-like modifiers, but has a different active site architecture and is truly specific for the cleavage of ubiquitin. We provide a detailed analysis of ZUFSP (Zn-finger and UFSP domain protein), the singular human member of this family, which contains multiple UBDs responsible for the specific action on K63-linked chains. By contrast, Mug105—a K48-preferring ZUFSP homolog from the fission yeast *Schizo-saccharomyces pombe*—lacks all UBDs and consists only of the core catalytic domain. A comparison of ZUFSP and Mug105 offers unique insights into the mechanism of how the evolutionary loss (or gain) of non-catalytic ubiquitin-binding domains can profoundly change the specificity of a deubiquitinase.

## Results

**ZUFSP and Mug105 are related to UFM1/Atg8 proteases**. The catalytic domain of the UFM1-specific protease (UFSP) family has been reported to be structurally related to that of the ATG4 family, enzymes that process the autophagy modifiers Atg8 and Atg12[11]. By performing sensitive sequence analysis using the generalized profile method[12], we found this structural similarity to be mirrored by a distant but highly significant sequence relationship ($p < 0.001$) between these families. Concurrently, a third protein family containing the uncharacterized human protein C6orf113/ZUFSP was found to exhibit significant sequence similarity ($p < 0.001$) to both UFSP and ATG4 families of cysteine proteases. As shown in Fig. 1a, the active site Cys and Asp residues of UFSP and ATG4 are conserved in members of the ZUFSP family, while the catalytic His is absent. Hypothesizing that ZUFSP might nevertheless be an active protease with a rearranged active site, we analyzed ZUFSP family members from a wide range of species and identified two highly conserved histidine residues, with His-491 being the best candidate for completing the active site of human ZUFSP (Supplementary Fig. 1). While the conserved catalytic domain is common to all

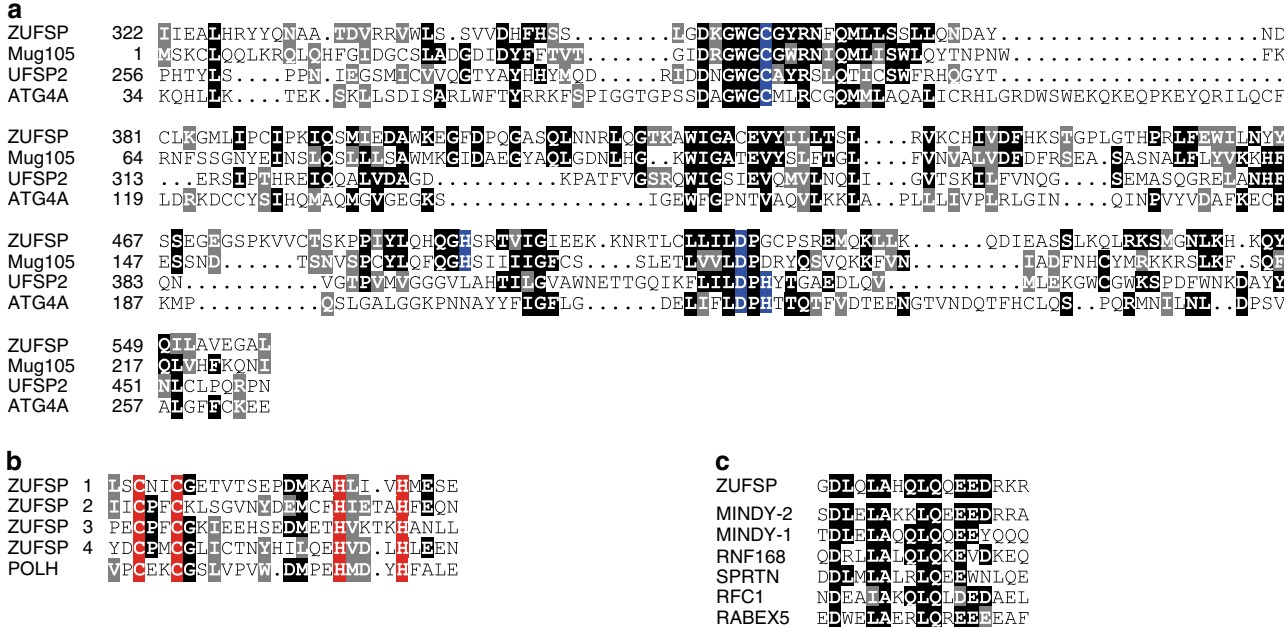

**Fig. 1** ZUFSP and Mug105 are related to UFM1/Atg8 proteases. **a** Structurally correct alignment of the catalytic domains of human ZUFSP (this work), mouse UFSP2 (3OQC) and human ATG4A (2P82). The *S. pombe* Mug105 sequence was added by sequence similarity to ZUFSP. Invariant and conservatively replaced residues are shown on black or gray background, respectively. Catalytic residues are highlighted in blue. **b** Conservation of the four UBZ-like zinc fingers of ZUFSP, in comparison to the structurally characterized UBZ finger (3WUP). **c** Conservation of the MIU domain of ZUFSP, in comparison to other human MIU domains

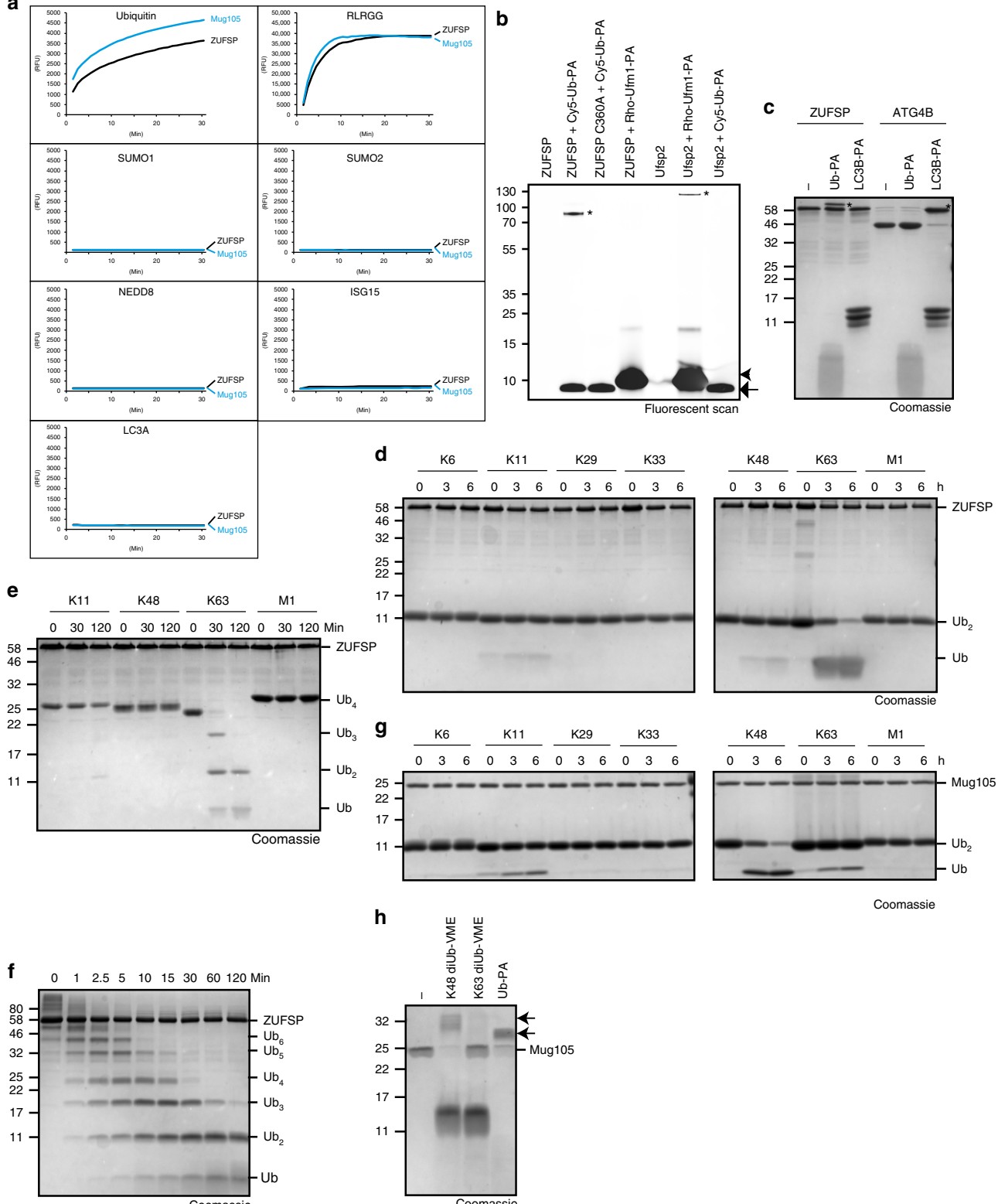

**Fig. 2** ZUFSP and Mug105 are ubiquitin-specific proteases with different chain specificity. **a** Activity assays with Ub-/UbL-AMC substrates shown as released fluorescence (RFU) over time (min) with ZUFSP or Mug105. Shown RFU values are the means of triplicates. **b** Fluorescent scan of a suicide probe reaction of ZUFSP or Ufsp2 with Cy5-Ub-PA (arrow) and Rho-Ufm1-PA (arrowhead). Asterisks (*) mark the shifted band after reaction. **c** Suicide probe reaction of ZUFSP or ATG4B with Ub-PA and LC3B-PA. Asterisks (*) mark the shifted band after reaction. **d**, **e** Linkage specificity analysis with ZUFSP. A panel of di-ubiquitin (**d**) or tetra-ubiquitin (**e**) chains was treated with ZUFSP for the indicated time points. **f** Time course of cleavage of K63-linked $Ub_{6+}$ chains by full-length ZUFSP. **g** Linkage specificity analysis with Mug105. A panel of $Ub_2$ was treated with Mug105 for the indicated time points. **h** Suicide probe reaction with Mug105 and K48-diUb-VME, K63-diUb-VME or Ub-PA. Arrows mark the shifted bands after reaction

members of the ZUFSP family, there are substantial differences in the non-catalytic regions. Some members, including Mug105 from the fission yeast *S. pombe*, consist solely of the catalytic domain, while most members contain one or more sequence regions with significant similarity to known ubiquitin-binding domains. Human ZUFSP contains four predicted C2H2 zinc fingers related to the ubiquitin-binding UBZ-class[13], followed by a predicted MIU (motif interacting with ubiquitin) region[14] (Fig. 1b, c). Thus, bioinformatical sequence analysis suggests that ZUFSP could be an active protease that is distantly related to various UBL-specific proteases, but has the potential to bind—and maybe cleave—polyubiquitin chains.

**ZUFSP and Mug105 are DUBs.** Since ZUFSP is related to UBL proteases and contains putative ubiquitin-binding domains, we tested bacterially expressed ZUFSP proteins for catalytic activity against ubiquitin and ubiquitin-related modifiers. Both human ZUFSP and the *S. pombe* homolog Mug105 were able to liberate the fluorophore AMC (7-amino-4-methylcoumarin) from a ubiquitin-AMC fusion, indicating a cleavage after the C-terminal Gly-76 of ubiquitin (Fig. 2a). The specific activities of ZUFSP and Mug105 against this substrate were 2.3 and 4.1 nmol substrate per mg enzyme per second, respectively (Supplementary Fig. 2a). By contrast, analogous AMC fusions of the ubiquitin-like modifiers SUMO1, SUMO2, NEDD8, ISG15 or LC3A were not processed (Fig. 2a). Surprisingly, both ZUFSP and Mug105 were highly active against RLRGG-AMC, a fusion of AMC, to a pentapeptide derived from the ubiquitin C terminus (Fig. 2a). This reaction follows the Michaelis–Menten kinetics with $K_M = 50.4\ \mu M$, $k_{cat} = 4.9\ s^{-1}$ for ZUFSP, and $K_M = 12.2\ \mu M$, $k_{cat} = 7.2\ s^{-1}$ for Mug105 (Supplementary Fig. 2b). The RLRGG-AMC peptide is not cleaved by typical DUBs, which require an intact ubiquitin moiety for activity, as shown here for USP21 (Supplementary Fig. 2c). While bacterially expressed ZUFSP did not react with C-terminally propargylated UFM1 (UFM1-PA, Fig. 2b) or the propargylated Atg8 homolog LC3B-PA (Fig. 2c), it readily reacted with propargylated ubiquitin (Ub-PA, Fig. 2b, c). Ub-PA is a covalent inhibitor that is highly selective for thiol DUBs[15]; the analogous inhibitor UFM1-PA reacts only with UFM1 proteases such as UFSP2, while LC3B-PA reacts with Atg8 proteases such as ATG4B (Fig. 2b, c).

When tested against a panel of di-ubiquitin species of different linkage types, ZUFSP showed a moderate activity towards K63-linked di-ubiquitin, with some minimal activity towards K11- and K48-linked species (Fig. 2d). When using tetra-ubiquitin substrates, ZUFSP showed K63-specific cleavage with a markedly increased activity: while K63-Ub$_4$ was completely hydrolyzed after 30 min (Fig. 2e), the hydrolysis of K63-Ub$_2$ was not completed after 6 h under comparable reaction conditions (Fig. 2d). This preference for Ub$_4$ over Ub$_2$ suggests an 'endo-cleavage' mode for ZUFSP, which became more evident when using longer K63-chain substrates: while the upper bands disappear within the first minutes, Ub$_2$ is remarkably inert and mono-ubiquitin appears late in the time course (Fig. 2f). The specificity for K63 cleavage is maintained in ZUFSP truncations: gradual shortening of the N-terminal region leads to decreased activity, but no change in the linkage preference (Supplementary Fig. 3a-c).

Unlike the K63-specific ZUFSP, the compact homolog Mug105 from *S. pombe* cleaved K48-linked di-ubiquitin better than other linkage types, with some residual activity against K63 and K11 species (Fig. 2g). A concordant observation was made when testing Mug105 with covalent DUB-specific inhibitors containing a reactive group between two ubiquitin units[16]. Mug105 reacted with the mono-ubiquitin-targeted probe Ub-PA and with the K48-targeted probe K48-diUb-VME, but not with the corresponding K63-targeted probe K63-diUb-VME (Fig. 2h). Thus,

both ZUFSP and Mug105 are linkage-specific DUBs, albeit with different specificities. The multitude of predicted ubiquitin-binding domains in ZUFSP, all absent from Mug105, might account for the strikingly different chain preference.

**ZUFSP and ATG4/UFSP structures differ in the active site.** For getting further insights into the unusual active site geometry of ZUFSP and the basis for its K63 specificity, we solved the crystal structure of a covalent complex between human ZUFSP (residues 232–578) and Ub-PA to a resolution of 1.7 Å (Table 1). The asymmetric unit contains one ZUFSP ubiquitin conjugate, which was almost completely resolved in the electron density. Only a short flexible loop (AA: 468–473) and eight amino acids at the very N terminus were not defined well enough in the electron density to permit reliable modeling. The ZUFSP fragment used for coupling to Ub-PA and subsequent crystallization starts before the predicted MIU domain and encompasses the conserved C-terminal region of human ZUFSP, including the catalytic core domain and adjacent elements predicted to be structured. The reaction product of thiol DUBs with Ub-PA resembles an intermediate stage of the protease reaction, in which the distal ubiquitin (poised for removal) occupies the S1 site of the enzyme. The ZUFSP structure forms a globular α/β-folded core with two prominent helical protrusions (Fig. 3a). The core

**Table 1 Data collection and refinement statistics**

| | ZUFSP S-SAD | ZUFSP native |
|---|---|---|
| *Data collection* | | |
| Space group | $P6_522$ | $P6_522$ |
| *Unit cell constants* | | |
| $a, b, c$ (Å) | 84.2, 84.2, 201.8 | 84.2, 84.2, 201.8 |
| $\alpha, \beta, \gamma$ (°) | 90, 90, 120 | 90, 90, 120 |
| Wavelength (Å) | 2.07 | 0.98 |
| Resolution (Å) | 59.13–2.3 (2.38–2.3) | 49.45–1.73 (1.79–1.73) |
| No. of observations | 4,098,044 (364,012) | 484,411 (48,504) |
| No. of unique reflections | 35,609 (3576)[a] | 44,866 (4337) |
| Multiplicity | 115.1 (101.8) | 10.8 (11.2) |
| Completeness (%) | 100 (100) | 100 (99) |
| $R_{merge}$ (%) | 11.7 (39.5) | 8.4 (82.4) |
| $R_{meas}$ (%) | 11.8 (39.7) | 14.3 (86.4) |
| $<I/\sigma(I)>$ | 47.7 (13.9) | 16.6 (2.5) |
| $CC_{1/2}$ (%) | 100 (99.7) | 99.8 (79.5) |
| Anomalous signal | | |
| CCano (59.13–3 Å) [3.08–3.0 Å]) (%) | 56 (31) | |
| $R_{pim}$ (59.13– 3 Å) [3.08–3.0 Å]) (%) | 0.8 (1.4) | |
| *Refinement* | | |
| Reflections used in refinement | | 44,862 |
| Number of test reflections | | 1930 |
| $R_{work}/R_{free}$ (%) | | 17.8/20.7 |
| Root-mean-square deviations | | |
| Bond lengths (Å) | | 0.005 |
| Bond angles (°) | | 0.6 |
| Average B factor (Å$^2$) | | |
| All macromolecule atoms | | 37.0 |
| Solvent molecules | | 39.5 |
| Other atoms | | 24.3 |
| Ramachandran plot (%) | | |
| Most favored | | 97.7 |
| Additionally allowed | | 2.2 |
| Disallowed | | 0.0 |

[a]Friedel pairs were kept separate

catalytic domain is clearly related to that of the UFM1-protease UFSP2 (PDB: 3OQC)[11]; the two structures can be superimposed with an RMS (root mean square deviation) distance of 3.65 Å over 200 residues (Fig. 3b and Supplementary Fig. 4a). A long 29-residue helix (α1) points away from the active site and provides the S1 binding surface for the outgoing distal ubiquitin—here occupied by the covalently coupled Ub-PA. The α2 and α3 helices form a hairpin-like structure protruding from the catalytic core in the opposite direction. This region has little contact with the remainder of the structure and might form the S1' site for binding the proximal (substrate) ubiquitin. Of particular interest is the organization of the active site (Fig. 3c). As expected from sequence analysis, the positions of the catalytic Cys-360 (start of α5) and Asp-512 (behind β5) are analogous to UFSPs and

ATG4[9,11]. However, the catalytic His-491 of ZUFSP is provided by the start of β4, whereas the catalytic histidine of UFSP and ATG4 proteases would have been expected between β5 and α10. In accordance with its catalytic role, His-491 is universally conserved within the ZUFSP family (Supplementary Fig. 1), and its essentiality for ZUFSP activity could be established (Supplementary Fig. 2d). Through the use of a different catalytic histidine, the active site geometry of ZUFSP resembles that of papain[17] and is almost a mirror image of the UFSP and ATG4 active sites (Fig. 3c). Both histidine positions appear equally suited for acidifying the catalytic cysteine. Nevertheless, a fundamental change in active site architecture, as observed here between the ZUFSP and UFSP families, is a rare event for evolutionary related proteases that have maintained similar activities. The oxyanion hole of ZUFSP has one remarkable difference to other cysteine proteases with similar active site architecture. In UFSPs and papain, the side chains of Tyr-282 and Gln-19 respectively provide polarized hydrogens for the electrophilic pocket. Mutating these polar residues abolishes the proteolytic activity[11,18]. In ZUFSP, the corresponding position is occupied by Ser-351, which cannot act as hydrogen donor due to its smaller side chain. Consequently, the S351A mutation showed no loss in activity (Supplementary Fig. 5a, b). Attempts to improve the catalytic rate by introducing more suitable side chains (S351Q or S351Y) proved unsuccessful (Supplementary Fig. 5a, b), further supporting the idea that the oxyanion hole in ZUFSP is different from that in UFSPs and papain.

**Contribution of the N terminus to ZUFSP catalytic activity.** While the crystal structure of ZUFSP in covalent complex with ubiquitin offers interesting insights into the catalytic mechanism and linkage specificity, the construct used for crystallization lacks the predicted UBZ-like zinc fingers. Moreover, the MIU domain, which forms the N-terminal part of the crystallized fragment, is not contacting the distal S1 ubiquitin. For assessing the contributions of the predicted UBDs to ZUFSP activity, a series of truncation mutants were analyzed in a time course experiment (Fig. 4a, b). When incubating full-length ZUFSP with long K63-linked ubiquitin chains ($Ub_{6+}$), most of the high-molecular-weight material was reduced to $Ub_6$ and shorter forms within 5 min, with very little mono-ubiquitin being generated. After 60 min, most of the substrate was present as di-ubiquitin, which—being a poor ZUFSP substrate—required several hours for further degradation. A ZUFSP construct starting at position 148, and thus lacking the first two zinc fingers, was slightly more active than full-length ZUFSP (Fig. 4b). By contrast, the construct used for crystallization, starting at position 232 before the MIU

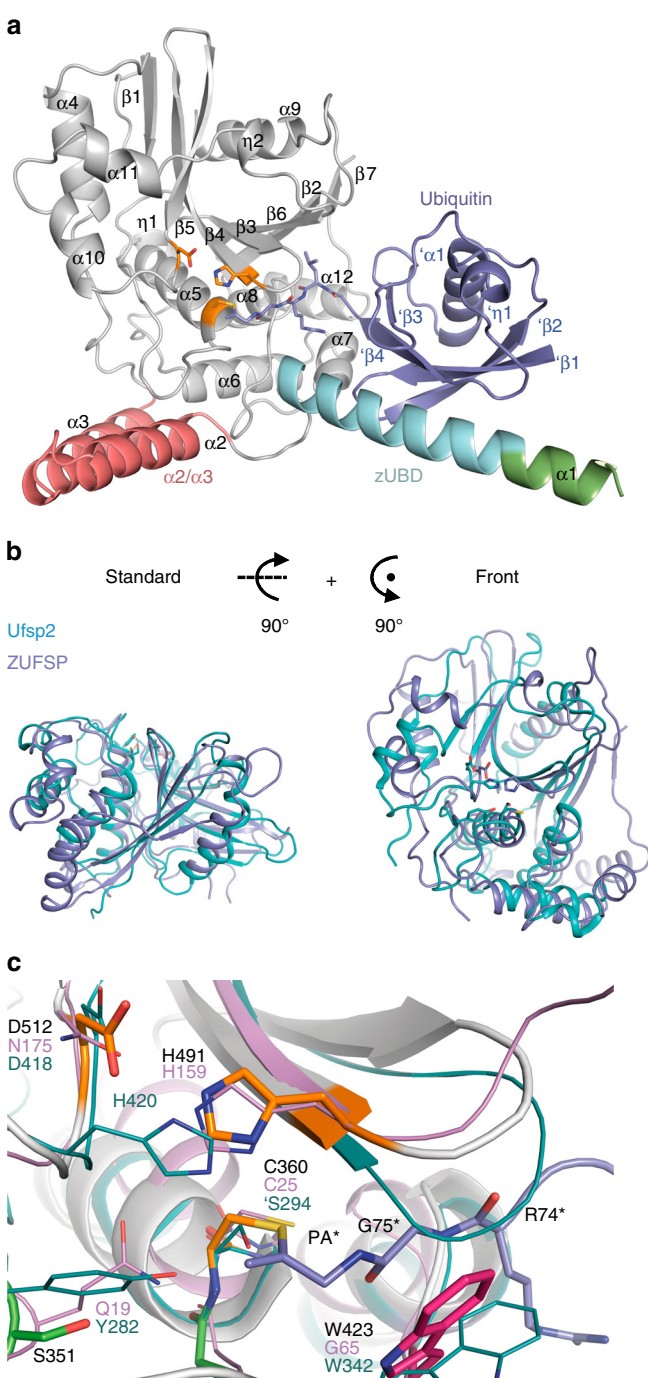

**Fig. 3** Crystal structure of ZUFSP$^{232-578}$ in covalent complex with Ub-PA. **a** Overview of the crystal structure in cartoon representation. The catalytic core of ZUFSP is shown in gray, ubiquitin in blue. The MIU region on helix α1 of ZUFSP is colored green, the zUBD region in cyan. The putative S1' ubiquitin-binding α2/α3 helices are shown in red. The catalytic triad is shown as sticks and colored orange. **b** Structural superposition of the catalytic domain of ZUFSP (blue) and UFSP2 (3OQC, cyan) in two perspectives. RMS distance is 3.65 Å over 200 residues. **c** Magnification of the active site of ZUFSP (gray). The catalytic triad is colored orange, putative components of the oxyanion hole in green and Trp-423, closing the substrate binding groove directly next to the active site, is in dark pink. Ubiquitin is shown in blue color. The active sites of Ufsp2 (cyan) and papain (violet) are superimposed. Structurally equivalent residues of Ufsp2 and ZUFSP are shown as sticks. Important residues of ZUFSP (black), Ufsp2 (cyan) and papain (violet) are labeled. Important ubiquitin residue labels contain asterisks. In the available structure (3OQC) of UFSP2, the catalytic cysteine was mutated to a serine, indicated here as 'S294

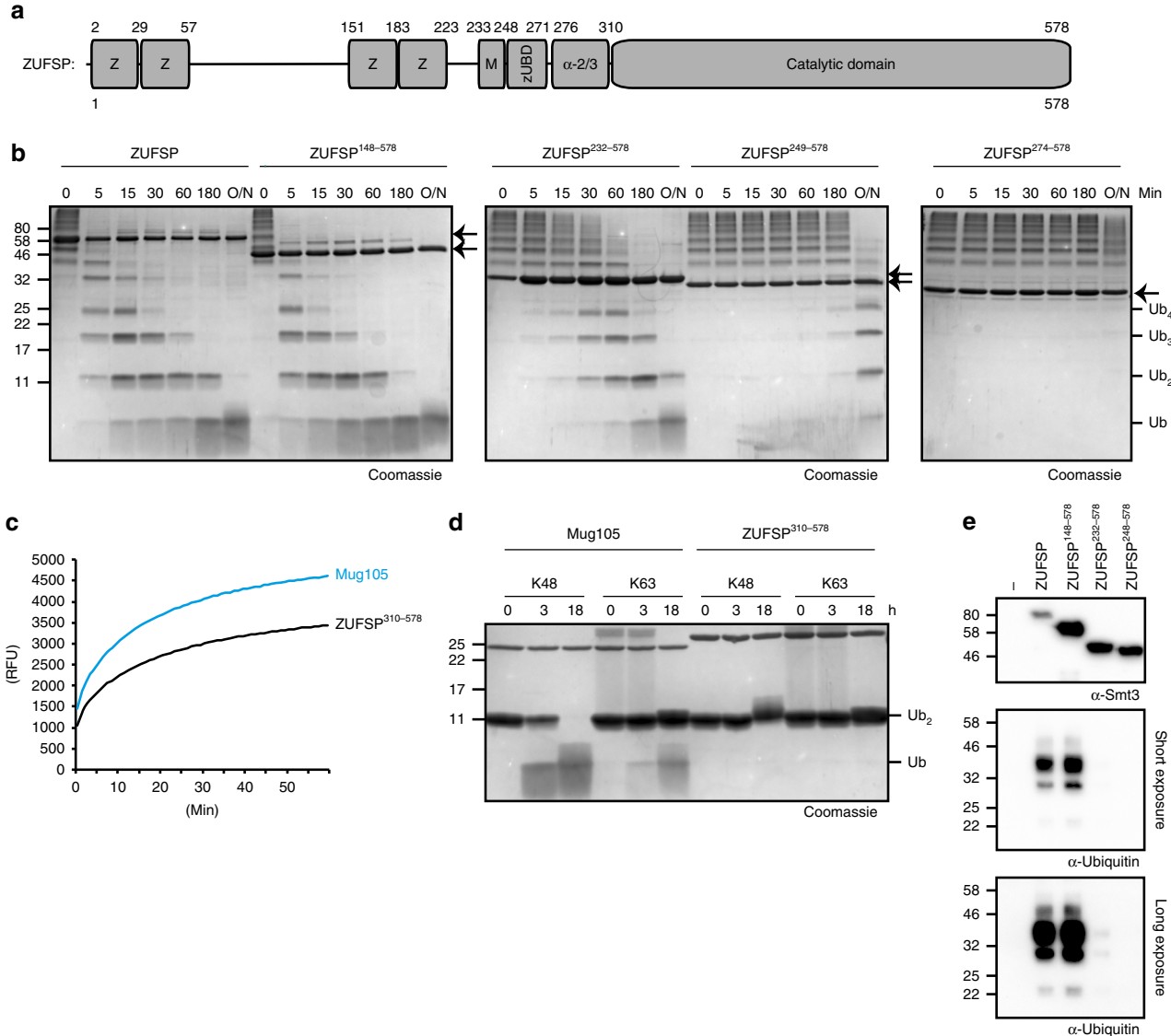

**Fig. 4** ZUFSP UBDs contribute to chain cleavage and specificity. **a** Schematic representation of ZUFSP domain architecture. UBZ-like zinc fingers (Z), MIU domain (M), novel ZUFSP ubiquitin-binding domain (zUBD) and α2/α3 region are shown as boxes. **b** Activity of ZUFSP FL and truncations lacking the UBDs against K63-linked $Ub_{6+}$ chains. Positions of the truncated ZUFSP proteins are indicated by arrows. **c** Comparison of Mug105 and ZUFSP[310-578] activity against ubiquitin-AMC. The shown RFU values are the mean of triplicates. **d** Chain specificity of ZUFSP catalytic core (ZUFSP[310-578]) compared to full-length Mug105. Both DUBs were tested against K48- and K63-linked $Ub_2$ for the indicated time points. **e** Pull-down analysis of ZUFSP (full-length and two N-terminal truncations) against a mixture of K63-linked $Ub_4$ and $Ub_5$ chains

domain, was substantially less active against K63 chains. Even shorter versions, such as the one starting at position 249 after the MIU, or starting at 274 after the α1 helix, were hardly active against K63 chains (Fig. 4b). The same trend was observed when testing the truncation mutants against $Ub_4$ and $Ub_2$ (Supplementary Fig. 3d, e).

The shortest fragment tested for activity starts at position 310 and corresponds to the core catalytic domain, analogous to the wild-type Mug105 protein. While both Mug105 and the human core fragment were active against ubiquitin-AMC (Fig. 4c) and thus properly folded, only Mug105 was able to cleave ubiquitin chains (Fig. 4d). Thus, the surface features conferring K48 specificity to Mug105 are not conserved in the human core fragment. The catalytic activity of the ZUFSP truncations is closely mirrored by the ability of the corresponding proteins to bind ubiquitin chains. In a pull-down experiment with immobilized His-tagged ZUFSP truncations and K63-liked ubiquitin chains, only the full-

length protein and the truncation starting at 148 showed robust ubiquitin-binding (Fig. 4e).

**Determinants of ZUFSP specificity.** Analysis of the contact surface between ZUFSP and the covalently bound ubiquitin revealed two major substrate recognition modalities. The interface most critical for proteolytic activity is formed by salt bridges involving two acidic residues of ZUFSP (Asp-406 and Glu-428) and the two arginine residues within the C-terminal tail of ubiquitin (Arg-72 and Arg-74) (Fig. 5a). These side-chain interactions position the ubiquitin C terminus next to the catalytic cysteine and are thus absolutely crucial for the cleavage of ubiquitin chains and the model substrate RLRGG-AMC (Fig. 5b, c). In both assays, mutation of either D406A or E428A renders ZUFSP as inactive as the active site mutant C360A. This Arg–Arg recognition, together with a narrow hydrophobic tunnel directly

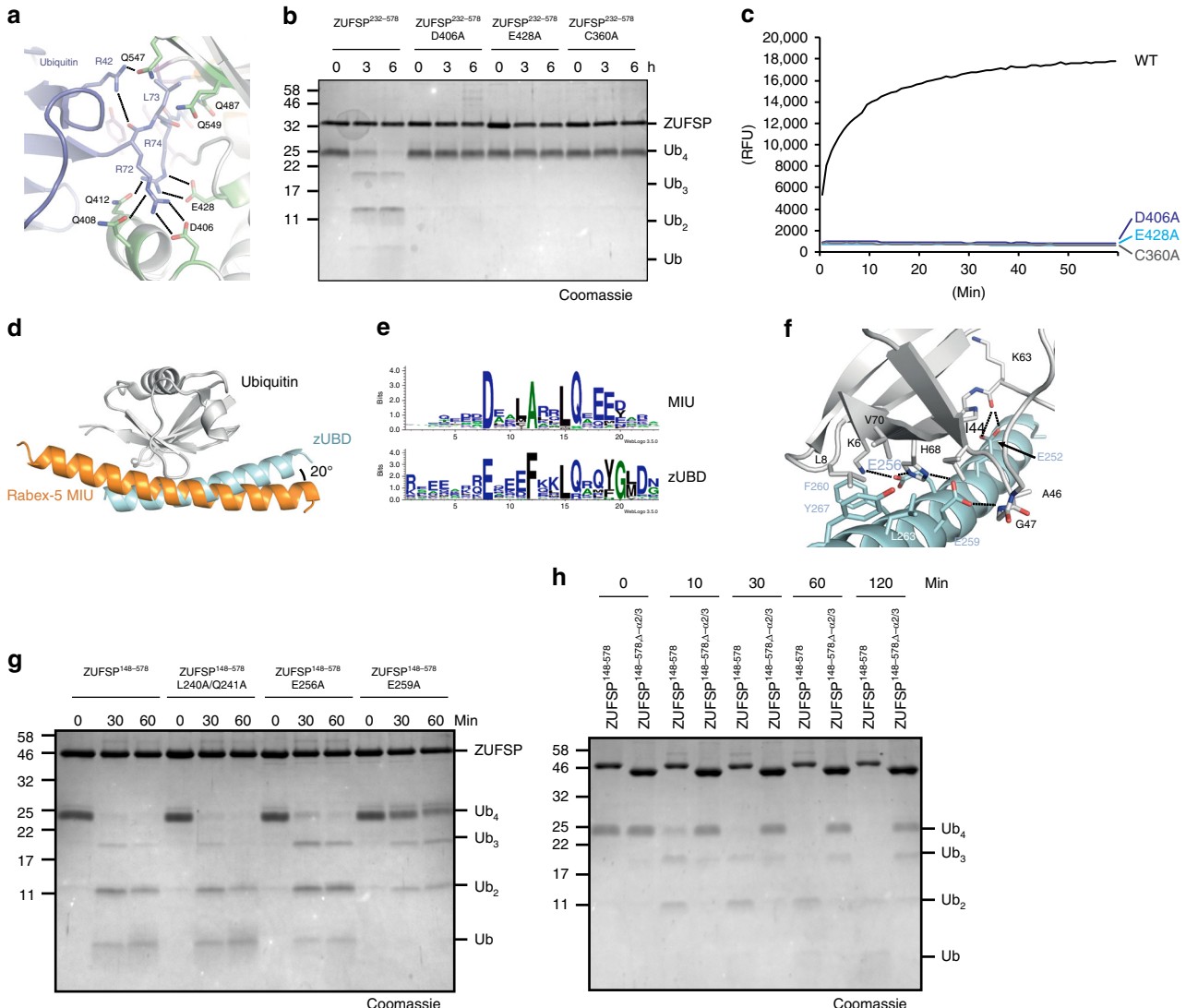

**Fig. 5** Determinants of chain specificity. **a** Recognition of ubiquitin C terminus by the catalytic core of ZUFSP. Ubiquitin (blue) and ZUFSP (gray/green) shown in cartoon representation with key residues highlighted as sticks. Blue and black residue labels refer to ubiquitin and ZUFSP, respectively. Salt bridges are indicated by dotted lines. **b** Activity of C-terminus recognition mutants (ZUFSP[232–578] D406A or E428A) against K63-linked Ub[4] was compared to ZUFSP[232–578] and inactive ZUFSP (ZUFSP[232–578] C360A). **c** Activity of mutants described in **b** against RLRGG-AMC. The RFU values shown are the means of triplicates. **d** Structural superposition of ubiquitin-binding interfaces to zUBD (cyan, this work) and the MIU domain of Rabex-5 (2FIF, orange). Orientation of the two helical ubiquitin-binding domains differ by 20°. **e** SeqLogo[45] representation of the consensus sequences for the MIU motif[14] (top) and the zUBD derived from the ZUFSP family as shown in Supplementary Fig. 1 (bottom). **f** Magnification of the interaction interface between ubiquitin and zUBD. Relevant residues are shown as sticks and labeled black in case of ubiquitin and cyan in case of zUBD. Electrostatic interactions are indicated as dotted lines. **g** Activity of a MIU mutant (ZUFSP[148–578] L240A/Q241A) and two zUBD mutants (ZUFSP[148–578]E256A and E259A) on K63-linked Ub[4], in comparison to wild-type ZUFSP[148–578]. **h** Activity time course of the α2/3-deletion mutant ZUFSP[148–578; Δ-α2/3] on K63-linked Ub[4] chains, compared to activity of the parental ZUFSP[148–578] construct

before the Ser-360, defines the specificity of ZUFSP for the C terminus of ubiquitin (RLRGG). The tunnel itself is formed by the core of the cysteine peptidase as well as the side chains of Tyr-267, Trp-423, Gln-489 and Gly-490. At least in the conformation found in our crystal structure, the space within the tunnel only permits glycine to pass (Supplementary Fig. 4b). A second interaction surface is formed by the second half of the ZUFSP α1 helix, which binds the Ile-44 patch of the S1ubiquitin in an orientation similar, but not identical to that of the Rabex-5 MIU domain (Fig. 5d). The ubiquitin-binding part of the α1 helix, here denoted as zUBD (ZUFSP ubiquitin-binding domain), contacts the Ile-44 patch of ubiquitin at an angle tilted by 20° relative to the MIU orientation. There is also a substantial difference in the sequence consensus (Fig. 5e), emphasizing

that—despite the similar binding mode–the zUBD is not just an unusual MIU. Close contacts between the zUBD and ubiquitin are formed by two acidic residues (Glu-256 and Glu-259) (Fig. 5f), which differ in their importance for ubiquitin cleavage: while the E259A mutant was hardly active against Ub[4], the E256A mutant showed only a modest reduction in activity (Fig. 5g). The N-terminal half of the ZUFSP α1 helix is formed by its canonical MIU domain, but is not bound to ubiquitin and only partially resolved in the available structure. Nevertheless, the canonical MIU is ideally positioned to bind a distal S2-ubiquitin K63-linked to the outgoing S1 ubiquitin present in the structure. A double mutant targeting the highly conserved Leu-240/Gln-241 residues of the canonical MIU motif was equally active as wild-type ZUFSP, suggesting that MIU activity is dispensable for chain

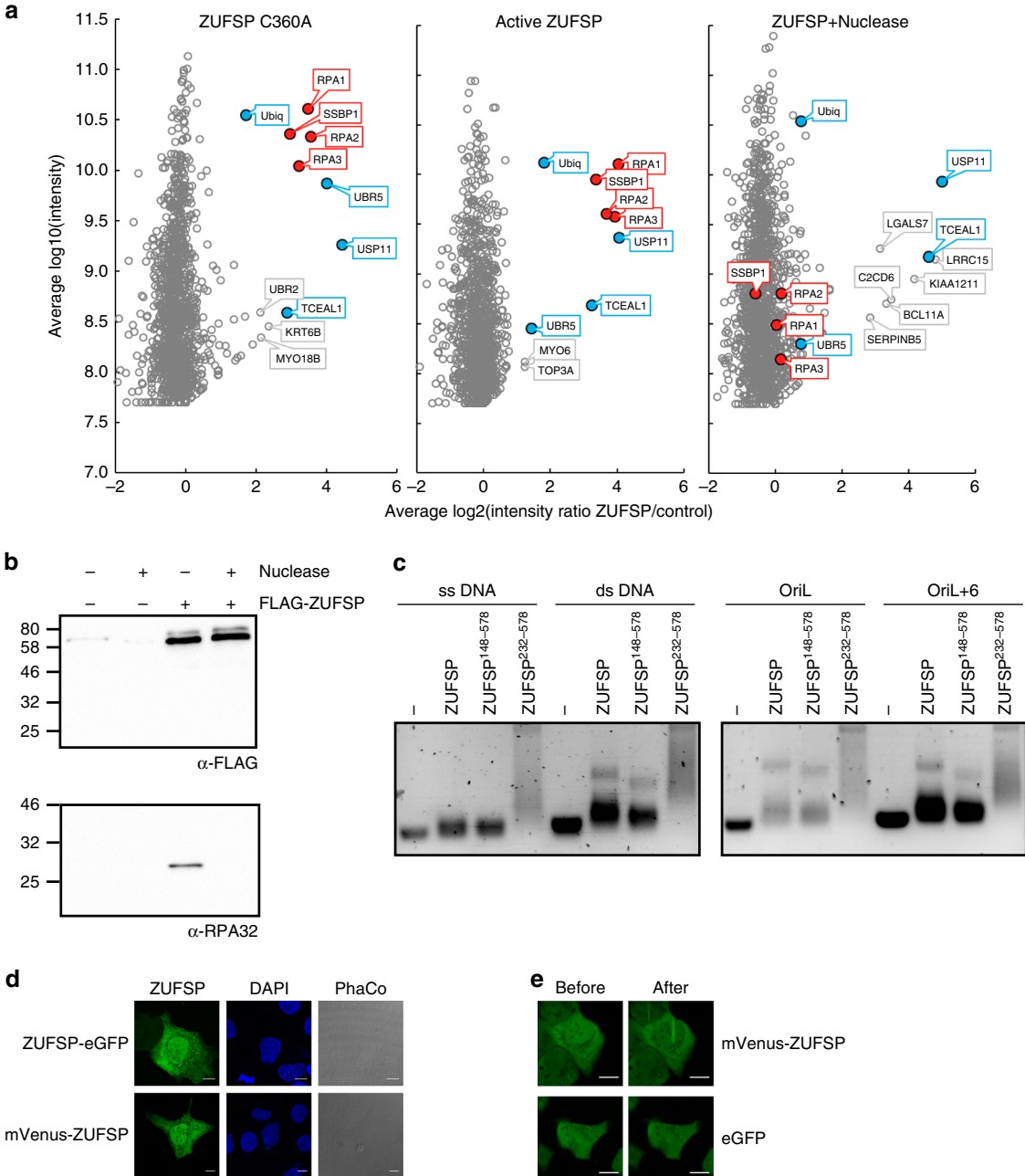

**Fig. 6** ZUFSP localization and interaction network. **a** FLAG-tagged versions of inactive ZUFSP[C360A] (left) and active ZUFSP (middle, right) were expressed in HEK293T cells and co-precipitating proteins quantified by mass spectrometry. Log2 enrichment ratios relative to uninduced/untransfected controls are plotted against log10 signal intensity. The right panel shows the results after nuclease treatment. Consistently enriched proteins are labeled in color, red for DNA-dependent and blue for DNA-independent enrichment. The bait ZUFSP is off-scale and hence not shown; the *x/y* coordinates are 9.7/11.1 (left), 11.0/11.2 (middle), and 10.5/11.6 (right panel). **b** FLAG-tagged ZUFSP was immunoprecipitated from HEK293T cells in the presence or absence of nuclease. Coimmunoprecipitated endogenous RPA32 was visualized with α-RPA32 4E4. **c** Electrophoretic mobility shift assay (EMSA) comparing the DNA-binding preferences of the full-length ZUFSP to the N-terminal truncations (ZUFSP[148–578] and ZUFSP[232–578]). All constructs were tested against a panel of oligonucleotides previously tested for SSBP1 binding[28], including ssDNA, dsDNA, short hairpin (OriL), and long hairpin (OriL+6). **d** Localization of ZUFSP N-terminally fused to mVenus or C-terminal fused to eGFP (green) was visualized in fixed U2OS cells. Cells are shown in phase contrast (PhaCo) and nuclei are stained with DAPI (blue). Scale bar = 10 μm. **e** Localization of mVenus-tagged ZUFSP to sites of NIR laser-induced DNA damage in U2OS cells (top panels), as compared to eGFP alone (bottom panels). Images were taken immediately before (left) and 10 s after 800 nm laser irradiation (right). Scale bar = 10 μm

cleavage (Fig. 5g). The positioning of the α2/α3 hairpin protrusion is expected to contact the proximal (S1') ubiquitin, which is not part of the available structure. A ZUFSP[Δ277-310] mutant lacking this region showed a strongly reduced activity against Ub₄ chains (Fig. 5h), suggesting a contribution of the α2/α3

region to ubiquitin recognition. All mutants tested in Fig. 5g, h remained fully active against the model substrate RLRGG-AMC (Supplementary Fig. 2e, f), suggesting that the observed difference in chain cleavage activity reflects substrate recognition rather than protein misfolding.

**ZUFSP localization and interaction network.** For getting insights into the biological function of ZUFSP, cellular interactors of this DUB were identified. To that end, FLAG-tagged full-length ZUFSP was stably expressed in HEK293T cells under tetracycline control. At 24 h after induction, the tagged ZUFSP construct was immunopurified together with its binding partners and analyzed by mass spectrometry. For comparison, a transiently transfected C360A mutant was immunopurified and analyzed accordingly. Relative to the uninduced and untransfected controls, both wild-type and inactive ZUFSP co-purified with a nearly identical set of highly enriched proteins, consisting of RPA1, RPA2, RPA3 (constituents of replication protein A (RPA)), SSBP1 (mito-chondrial replication protein), the deubiquitinase USP11, the ubiquitin ligase UBR5, ubiquitin itself, and the uncharacterized protein TCEAL1 (Fig. 6a). Since the RPA proteins and SSBP1 are known to bind single-stranded DNA (ssDNA), we tested the interactions for DNA dependence by repeating the co-purification in the presence of nuclease. After removal of DNA, all four ssDNA binding proteins were no longer enriched, while binding to USP11, TCEAL1, and ubiquitin was not affected. The DNA-dependent binding of ZUFSP to the RPA complex was confirmed by co-immunoprecipitation experiments. Full-length ZUFSP expressed in HEK293T cells co-precipitated endogenous RPA2, as visualized by an α-RPA32 antibody, while no RPA2 co-precipitation was observed in the presence of nuclease (Fig. 6b). To test whether ZUFSP is itself an ssDNA binding protein, electrophoretic mobility shift assays (EMSAs) were performed. Full-length ZUFSP was able to partially shift ssDNA, hairpin and double-stranded DNA (dsDNA) oligonucleotides, while the 232–578 truncation was a better binder (Fig. 6c). The poor binding of full-length ZUFSP and the preference for structures containing dsDNA makes it unlikely that a direct ssDNA binding accounts for the DNA-dependent interaction with RPA subunits and SSBP1. Since available ZUFSP antibodies failed to stain endogenous ZUFSP in immunofluorescence experiments, the subcellular localization in U2OS cells was determined for ectopically expressed ZUFSP fused to fluorescent proteins at either N or C terminus. Both constructs showed a uniform distribution throughout cytoplasm and nucleus (Fig. 6d). Two-photon near-infrared (NIR) laser microirradiation of U2OS nuclei lead to a partial and short-lived (<1 min) recruitment of ZUFSP to irradiated areas (Fig. 6e).

## Discussion

The discovery that C6orf113/ZUFSP and its homologs form a seventh class of deubiquitinating enzymes is interesting in several respects, including DUB evolution, catalytic mechanism, and the role of ubiquitin-binding domains in determining DUB specificity. The existence of—so far—seven different DUB classes also raises the question of how the different deubiquitination tasks are distributed between the different classes.

In sequence databases, the previously uncharacterized protein C6orf113 is referred to as ZUFSP (zinc finger with UFM1-specific peptidase domain protein) because in the Pfam database[19] its sequence scores significantly against a Hidden Markov Model (PF07910) describing the UFM1 proteases. However, depending on the exact sequence search method and parameters, the ZUFSP family appears to be nearly equidistant to the UFSP and ATG4 families, with a distance resembling that between UFSP and ATG4 (Supplementary Fig. 6). Thus, ZUFSP is clearly not a member of the UFSP family, but rather forms a superfamily with the processing enzymes for UFM1 and Atg8/Atg12. Both UFM1 and Atg8/Atg12 are extremely divergent members of the ubiquitin-like modifier family with no overt sequence similarity to ubiquitin. Unlike the canonical 'GG' motif found at the C

terminus of most ubiquitin-like modifiers, UFM1 and Atg8/Atg12 have only the second glycine residue conserved. Considering the high divergence between the three modifier types, it was surprising that one superfamily member is a linkage-specific endo-cleaving ubiquitin isopeptidase. A possible explanation for the substrate switch might be the different active site geometry, in combination with the presence of multiple UBDs that determine cleavage specificity within the ZUFSP/Mug105 family.

In evolutionary terms, the ZUFSP family is rather ancient, with recognizable members from all eukaryotic kingdoms. However, the phyletic distribution is characterized by many independent gene loss events, suggesting that ZUFSP function is either not universally required or that another DUB has been co-opted in certain lineages. ZUFSP family members are widespread in animals, plants, and fungi; they are also found in several other taxa, including *Cryptophyta*, *Alveolata*, *Amoebozoa*, and *Rhizaria*. Gene loss events affect many common model organisms: unlike some other nematodes, *Caenorhabditis elegans* lacks a ZUFSP-like protein; the same is true for the insect model *Drosophila melanogaster* and the budding yeast *Saccharomyces cerevisiae*, which both lack this DUB despite clear ZUFSP homologs being present in other *Dipterans* and *Ascomycetes*. This heterogeneity is also seen in the surprisingly diverse domain architectures of ZUFSP family members. All family members share the conserved catalytic domain anchored to the C terminus. Several protists have minimalistic proteins resembling Mug105, while the N-terminal regions of most taxa contain at least one ubiquitin-binding domain. Recurring architectures include 1xUBZ (most plants), 2xUBZ (most ascomycete fungi), 5xUBZ and 1xzUBD (insects), or 3xMIU but no zinc fingers in several alveolates (Supplementary Fig. 7). The UFSP proteases have also been lost in many lineages, together with the entire UFM1-modification system, but there is little correspondence to the species lacking ZUFSP family proteases. The absence of UFM1 and UFSPs from the yeast *S. pombe* underscores once more that Mug105 has no connection to UFM1 signaling. Despite their overall similarity in structure and sequence, ZUFSP proteases use an active site histidine residue different from that used in the UFSP and ATG4 families. Apart from the active site residues, all three families show a structural fold similar to the papain-type proteases, a fold that is also used by most other DUBs and UBL proteases[20]. In fact, the active site arrangement of ZUFSP, with the catalytic histidine upstream of the catalytic aspartate (C-H-D), is typical of papain fold proteases including USP-type deubiquitinases. It is therefore likely that the UFSP and ATG4 families evolved from a ZUFSP-like ancestor, and that the C-D-H active site of the two UBL-protease families is a derived feature.

The globular portion of ZUFSP, which is resolved in the covalent complex structure, has two regions of contact with the distal (S1) ubiquitin, which—at least partially—explains the specificity for K63-linked ubiquitin chains. The recognition of the two Arg residues in the ubiquitin C terminus through salt bridges is not unusual for DUBs. A similar arrangement can be seen in the substrate complex of USP21 and other USP-type deubiquitinases[21]. The resulting interactions position the ubiquitin C terminus favorably relative to the active site and should help in the recognition of the sequence R-x-R-G-G. Since mutating only one of the two acidic recognition counterparts is sufficient to abrogate the activity, recognition of both C-terminal arginine residues is required. Accordingly, the peptide-based model substrate RLRGG-AMC is cleaved very effectively, while NEDD8 (ending on ALRGG), SUMO1 (EQTGG), and SUMO2 (QQTGG) are not processed. However, ISG15 also ends on RLRGG but is not recognized by ZUFSP, showing that a second recognition layer must be in place. Contact site analysis suggests that the binding of the newly defined zUBD domain to the Ile-44-patch of

the S1 ubiquitin is responsible for the second recognition event, most likely enhanced by binding of the adjacent MIU domain to the S2-ubiquitin, which is not part of the structure. The zUBD domain binds ubiquitin in an approximately MIU-like manner, albeit with a different angle and a different consensus sequence for the contacting residues. The combination of zUBD and MIU on a contiguous helix forms a suitable surface for the recognition of K63 linkages, analogous to the recognition of K63 chains by the tandem-UIM domains of Rap80[22]. The available structure does not reveal how this second binding interface excludes ISG15, a ubiquitin fold with the proper RLRGG C terminus but without Ile-44 patch, from being cleaved by ZUFSP. One attractive model would posit that in the absence of ubiquitin, the α1-helix would fold back and obstruct access to the active site—at least for substrates larger than the RLRGG-AMC pentapeptide. On the other hand, the α1-deficient ZUFSP[274–578] variant is severely defective for ubiquitin chain cleavage, suggesting that the zUBD/MIU binding to the Ile-44 patch, together with hydrogen bonds formed by the ubiquitin-specific Arg-42 residue (Fig. 5a), might just be a necessary contribution to the overall substrate affinity.

Particularly intriguing is the different linkage specificity of human ZUFSP and its *S. pombe* homolog Mug105. While in ZUFSP, the K63 specificity can be largely rationalized by the rigid positioning of recognizable UBDs relative to the active site, the lack of a Mug105 crystal structure precludes the identification of the S1 and S1' sites required for K48 specificity. Since Mug105 is a radically reduced version of the ZUFSP architecture, it is safe to assume that the substrate recognition sites of Mug105 are found on surfaces that are not accessible in the human ZUFSP protein. A truncation of human ZUFSP[310-578], down to the core peptidase domain as found in Mug105, did not recapitulate the shift to K48 specificity, but rather abrogated chain cleavage altogether. Apparently, the surface required for K48 recognition is not conserved in the human protein. While the maximally shortened ZUFSP[310–578] no longer cleaved ubiquitin chains, it was still active against ubiquitin-AMC and RLRGG-AMC, showing that the truncated protein is properly folded and has maintained a functional active site. In summary, human ZUFSP has a unique modular architecture, consisting of a catalytic core made for cleaving after R-x-R-G-G motifs, but not able to efficiently recruit ubiquitin chains to the active site. The necessary capability for ubiquitin recruitment and linkage specificity is conferred to the core domain by a number of ubiquitin-binding domains, such as the rigidly linked zUBD and MIU, as well as the flexibly linked UBZ-like domains. It will be interesting to study if the various UBD classes linked to the catalytic core of other ZUFSP family members might confer yet different linkage specificities.

Despite their different domain architecture, human ZUFSP and *S. pombe* Mug105 appear to be orthologs, although it cannot be formally ruled out that both metazoan and fungal lineages have experienced gene losses of the true orthologs, leaving ZUFSP and Mug105 as pseudo-orthologs[23]. In either case, the difference in linkage specificity implies that ZUFSP and Mug105 have either assumed different biological roles or work in a biological pathway, which in humans and *S. pombe* is governed by different chain types. Neither ZUFSP nor Mug105 are functionally characterized. The only fact reported about Mug105 is its transcriptional upregulation during meiosis[24], which is not informative since several other meiotically upregulated genes in *S. pombe* are involved in pathways without obvious connections to meiosis.

Our interaction studies in HEK293T cells found the three subunits of RPA as the most enriched interactors, along with the mitochondrial ssDNA binding protein SSBP1. This interaction is corroborated by a recently reported incidental finding of ZUFSP in a screen for RPA interactors[25]. The co-precipitation of endogenous RPA with ectopically expressed ZUFSP further supports

the authenticity of this interaction. RPA and SSBP1 are known to perform analogous roles during replication and homologous recombination in the nucleus and mitochondria, respectively. During meiotic recombination and recombination repair, the two factors perform a similar role in stabilizing the ssDNA of the displacement loop (D-loop), which arises during strand invasion[26,27]. Due to their different localization, it is unlikely that RPA and SSBP1 form a complex, suggesting that ZUFSP recognizes these two similar factors independently. Both SSBP1 and RPA1 are known to bind ssDNA, which raises the question of whether their interaction with ZUFSP is direct or possibly bridged by ssDNA. Indeed, when repeating the interaction experiment in the presence of a non-specific bacterial nuclease, the enrichment of SSBP1 or RPA components was lost, while other interaction partners such as USP11 and TCEAL1 were not affected. These findings suggest that ZUFSP either recognizes a nucleic acid containing complex of RPA and SSBP1 or that ZUFSP itself is a DNA-binding protein. As seen in the EMSA experiments, ZUFSPs do in fact bind to DNA, but several observations make it unlikely that ZUFSP DNA-binding properties can explain the DNA-dependent association to RPA and SSBP1. When comparing the DNA-binding profile of ZUFSP to that of SSBP1 using the same set of oligonucleotides[28], it becomes apparent that full-length ZUFSP binds DNA rather poorly and prefers dsDNA-containing binding partners over pure ssDNA partners. By contrast, SSBP1 does not bind to dsDNA and prefers the ssDNA oligonucleotide over those containing hairpin structures[28]. Thus, a more complex recognition mode of the replication and recombination factors RPA and SSBP1 has to be assumed.

Summing up, the highly reproducible and DNA-dependent interaction of ZUFSP with nuclear and mitochondrial replication factors suggests a role of this DUB class in the regulation of replication and/or homologous recombination. This idea is compatible with the presence of ZUFSP in the nucleus and its transient recruitment to NIR-induced DNA damage sites. The biochemical properties of ZUFSP suggest that it generally targets K63 chains, rather than particular substrates, after being recruited to its site of action by one or more of its interaction partners. Ubiquitination by K63-linked chains are a hallmark of several DNA damage pathways[29,30]. While the processes activating the damage-responsive ubiquitin ligases are reasonably well understood, the removal of K63 chains during or after resolution of the damage is more enigmatic. The discovery of ZUFSP as a K63-DUB, making DNA-mediated interactions with RPA and SSBP1, will be instrumental for addressing these important questions.

## Methods

**Constructs and cloning**. ZUFSP was cloned from HEK293 complementary DNA and Mug105 from *S. pombe* genomic DNA (kind gift from J. Dohmen, University of Cologne) using Phusion DNA Polymerase (New England Biolabs). For protein purification, all constructs were cloned in the pOPIN-S vector[31] using the In-Fusion® HD cloning system (Takara Clontech). Point mutations were generated using the QuikChange Lightning kit (Agilent Technologies). For the generation of the stable cell line full-length ZUFSP was cloned into the pCDNA5/FRT/TO vector (ThermoFisher). ZUFSP was cloned into popinE-3C-eGFP (kind gift from Ray Owens, OPPF UK) and pCL-Neo-mVENUS (gift from Niels Gehring, University of Cologne) for localization studies. Constructs for ubiquitin-PA purification (pTXB1-ubiquitin[1–75]) and pOPIN-S USP21[196-565] were a kind gift of D. Komander (MRC LMB Cambridge).

**Protein purification**. Full-length ZUFSP, all truncations and mutants, and Mug105 were expressed from pOPIN-S vector with an N-terminal 6His-Smt3-tag. All constructs were transformed into *Escherichia coli* (Strain: Rosetta (DE3)pLysS). Then, 6–12 l cultures were grown in LB media at 37 °C until the OD600 of 0.8 was reached. The cultures were cooled down to 18 °C and protein expression was induced by addition of 0.2 mM isopropyl β-D-1-thiogalactopyranoside (IPTG). For full-length ZUFSP and ZUFSP[148-578] constructs, 0.1 mM ZnSO4 was added in addition to the IPTG. After 16 h, the cultures were harvested by centrifugation at

$5000 \times g$ for 15 min. After freeze thaw, the pellets were resuspended in binding buffer (300 mM NaCl, 20 mM TRIS pH 7, 20 mM imidazole, 2 mM β-mercaptoethanol) containing DNase and Lysozyme, and lysed by sonication. Lysates were clarified by centrifugation at $50,000 \times g$ for 1 h at 4 °C and supernatant was used for affinity purification on HisTrap FF columns (GE Healthcare) according to the manufacturer's instructions. For all constructs, except the ones used for pull-down analysis, the 6His-Smt3 tag was removed by incubation with Senp1[415-644] and concurrent dialysis in binding buffer. The liberated affinity-tag and Senp1 were removed by a second round of affinity purification with HisTrap FF columns (GE Healthcare). If necessary, proteins were further purified by anion exchange chromatography (HiScreen Q HP, GE Healthcare) or cation exchange chromatography (HiScreen SP HP, GE Healthcare). All proteins were finally subjected to size exclusion chromatography (HiLoad 16/600 Superdex 75 or 200 pg) in 20 mM TRIS pH 7, 150 mM NaCl, 2 mM dithiothreitol (DTT), concentrated using VIVASPIN 20 Columns (Sartorius), flash frozen in liquid nitrogen, and stored at −80 °C.

MBP-Ufsp2 was expressed and purified as described previously[32]. In brief, MBP-Ufsp2 was expressed as indicated above. Bacteria were lysed in binding buffer (20 mM TRIS pH 7.5, 200 mM NaCl, 2 mM DTT) and the supernatant was used for affinity purification on a MBPTrap (GE Healthcare). MBP-Ufsp2 was eluted with binding buffer containing 10 mM maltose, subjected to size exclusion chromatography (HiLoad 16/600 Superdex 200 pg; GE Healthcare) in 20 mM TRIS pH 7.5, 150 mM NaCl, 2 mM DTT, concentrated using VIVASPIN 20 Columns (Sartorius), flash frozen in liquid nitrogen, and stored at −80 °C.

**AMC activity assays**. Activity assays of DUBs against AMC-labeled Ub/UbL substrates were performed using reaction buffer (150 mM NaCl, 20 mM TRIS pH 7, 10 mM DTT) 1 μM DUBs and 10 μM zRLRGG-AMC (BACHEM AG, Switzerland), 1 μM Sumo1-AMC, 1 μM Sumo2-AMC (Boston Biochem, Inc., USA), 1 μM ISG15-AMC (Boston Biochem, Inc., USA), 1 μM Nedd8-AMC (ENZO Life Sciences GmbH, Germany), LC3A-AMC (Boston Biochem, Inc., USA), or 5 μM Ub-AMC (UbiQ-Bio, The Netherlands). The reaction was performed in black 96-well plates (Corning) at 30 °C and released fluorescence was measured using the Infinite F200 Pro plate reader (Tecan) equipped for excitation wavelength of 360 nm and an emission wavelength of 465 nm. The measurements were performed in triplicate and the mean is presented.

**Kinetics**. For determination of the specific activity against Ub-AMC the initial velocity was determined from a measurement of 100 nM DUB against 5 μM Ub-AMC in reaction buffer (150 mM NaCl, 20 mM TRIS pH 7, 10 mM DTT). Steady-state kinetics of ZUFSP or Mug105 against RLRGG-AMC were measured in reactions containing 100 nM DUB and the indicated concentrations of RLRGG-AMC in reaction buffer. Measurements were performed at 30 °C in triplicate. Initial velocities were plotted against the RLRGG-AMC concentrations and fitted to the Michaelis–Menten equation using Prism 6 (GraphPad) software.

**Ub and LC3B-PA synthesis**. The constructs pTXB1-ubiquitin[1–75] pTXB1-LC3B[1–119] were used to express ubiquitin or LC3B as a C-terminal intein fusion protein as described in ref.[33]. In brief, the fusion protein was affinity purified in buffer A (20 mM Hepes, 50 mM sodium acetate, pH 6.5, 75 mM NaCl) from clarified lysates using Chitin Resin (New England Biolabs) following the manufacturer's protocol. On-bead cleavage was performed by incubation with cleavage buffer (buffer A containing 100 mM MesNa (sodium 2-mercaptoethanesulfonate)) for 24 h at room temperature (RT). Resin was washed extensively with buffer A and pooled fractions were concentrated and subjected to size exclusion chromatography (HiLoad 16/600 Superdex 75) with buffer A. To synthesize Ub/LC3B-PA, 300 μM Ub/LC3B-MesNa were reacted with 600 μM propargylamine hydrochloride (Sigma Aldrich) in buffer A containing 150 mM NaOH for 3 h at RT. Unreacted propargylamine was removed by size exclusion chromatography and Ub/LC3B-PA was concentrated using VIVASPIN 20 Columns (3 kDa cutoff, Sartorius) flash frozen and stored at −80 °C.

**Rho-UFM1-PA synthesis**. UFM1 was synthesized by total linear solid-phase peptide synthesis (SPPS) on a Syro II MultiSyntech Automated Peptide synthesizer using standard 9-fluorenylmethoxycarbonyl (Fmoc) based solid-phase peptide chemistry at a 40 μmol scale, using fourfold excess of amino acids relative to preloaded Fmoc amino acid trityl resin (0.2 mmol/g, Rapp Polymere GmbH). Peptide couplings were performed using benzotriazol-1-yl-oxytripyrrolidinophosphonium hexafluorophosphate (PyBOP, 4 equivalent (equiv)) and N,N-diisopropylethylamine (DiPEA, 8 equiv) in N-methyl-2-pyrrolidone (NMP) for 45 min. Fmoc removal was executed using 20% piperidine in NMP for 2 × 2 and 1 × 5 min. 5-Carboxy-Rhodamine-110 was coupled to the N terminus of resin-bound UFM1 and subsequently the fluorescently labeled UFM1 was cleaved off the resin using hexafluoroisopropanol (HFIP) in dichloromethane (DCM) (1:4 v/v) for 2 times for 20 min and filtered, thereby only liberating the C-terminal carboxylic acid while leaving all other protective groups in place. The flow-through was collected and concentrated in vacuo, followed by coevaporation with dichloroethane (3×) to remove residual HFIP. Subsequently, the protected peptide was dissolved in DCM and reacted with PyBOP (5 equiv), DiPEA (15 equiv), and propargylamine (15 equiv) for 16 h. The reaction was concentrated in vacuo and treated with 90.5%

trifluoroacetic acid, 5% water, 2.5% phenol, and 2% tri-isopropylsilane for 2.5 h to globally remove all protective groups. The fully deprotected peptide was precipitated from Et2O/Pentane (1:1, v/v) and subsequently redissolved in dimethyl sulfoxide/water (1:9, v/v) and purified using reverse-phase high-performance liquid chromatography. Lyophilisation of the appropriate fractions yielded the target activity-based probe, which was analyzed by liquid chromatography–mass spectrometry (LC-MS; Waters 2795 Separation Module (Alliance HT), Waters 2996 Photodiode Array Detector (190–750 nm), Phenomenex Kinetex C18 (2.1 × 100, 2.6 μm) column, and LCT$^{TM}$ orthogonal acceleration time of flight mass spectrometer.

**Suicide probe assay**. DUBs were prediluted to 2× concentration (10 μM) in reaction buffer (20 mM TRIS pH 7, 150 mM NaCl and 10 mM DTT) and 1:1 combined with 100 μM Ub-PA, LC3B-PA, 2K48-VME (UbiQ-Bio), or 2K63-VME (UbiQ-Bio). After 16 h of incubation at 4 °C, the reaction was stopped by addition of Laemmli buffer, resolved by sodium dodecyl sulfate–polyacrylamide gel electrophoresis (SDS-PAGE), and Coomassie stained. For demonstrating Ub specificity of ZUFSP, full-length ZUFSP was incubated with either Cy5-Ub-Propargylamine[15] (Cy5-Ub-PA) (10 μM) or Rho-UFM1-Propargylamine (Rho-Ufm-PA) (10 μM) in 50 mM Tris-HCl pH 7.5, 100 mM NaCl, and 2 mM DTT at 37 °C for 30 min. The reaction was quenched by the addition of reducing sample buffer and heating at 95 °C for 3 min. Samples were resolved by SDS-PAGE and labeled enzymes were visualized by in-gel fluorescence scanning using the Typhoon FLA imaging system (GE Healthcare Life Sciences) ($\lambda$ex/ $\lambda$em = 625/ 680 nm and $\lambda_{ex}$/ $\lambda_{em}$ = 480/ 530 nm) and subsequent silver staining.

**Chain generation**. Met1-linked di-ubiquitin was expressed as a linear fusion protein and purified by ion exchange chromatography and size exclusion chromatography. K11-, K48-, and K63-linked ubiquitin were enzymatically assembled using UBE2SΔC (K11), CDC34 (K48), and Ubc13/UBE2V1 (K63) as previously described[34,35]. In brief, ubiquitin chains were generated by incubation of 1 μM E1, 25 μM of the respective E2 and 2 mM ubiquitin in reaction buffer (10 mM ATP, 40 mM TRIS (pH 7.5), 10 mM MgCl2, 1 mM DTT) for 18 h at RT. The reaction was stopped by 20-fold dilution in 50 mM sodium acetate (pH 4.5) and chains of different lengths were separated by cation exchange using a Resource S column (GE Healthcare). Elution of different chain lengths was achieved with a gradient from 0 to 600 mM NaCl.

**Chain cleavage assays**. DUBs were preincubated in 150 mM NaCl, 20 mM TRIS pH 7, and 10 mM DTT for 10 min. The cleavage was performed for the indicated time points with 5 μM DUBs and either 25 μM di-ubiquitin (K11, K63, and K48 synthesized as described above, others from Boston Biochem) or 5 μM tetra-ubiquitin (Boston Biochem) at RT, stopped with Laemmli buffer, resolved by SDS-PAGE, and Coomassie stained.

**Crystallization**. 100 μM ZUFSP$^{232-578}$ was incubated with 200 μM ubiquitin-PA for 18 h at 4 °C. Unreacted ZUFSP and Ub-PA were removed by size exclusion chromatography. The covalent ZUFSP$^{232-578}$ and Ub-PA complex (12 mg/ml) was crystallized using the vapor diffusion sitting drop method. Crystallization trials were set up with drop ratios of 1:2, 1:1, 2:1 protein solution to precipitant solution with a total volume of 300 nl. Initial crystals appeared in PEG/Ion (Hampton Research) E4 (0.2 M sodium malonate pH 5, 20% polyethylene glycol 3350 (PEG 3350)) after 2 days at 4 °C. Optimization was carried out with 3 μl drops (protein/precipitant ratios: 2:1, 1:1 and 1:2) and precipitant solutions varying in pH or PEG 3350 concentration respectively. Best crystals were obtained from the crystallization trial where protein solution was mixed 1:1 with precipitant solution composed of 0.2 M sodium malonate pH 5, 20% PEG 3350. Crystals were flash-cooled in reservoir solution containing 20 % (v/v) glycerol.

**Data collection, phasing, model building, and refinement**. Diffraction data were collected at beamline P13 at EMBL Hamburg, Deutsches Elektronen-Synchrotron (DESY), Hamburg, Germany, and processed using X-ray detector software (XDS)[36]. For S-SAD (single-wavelength anomalous dispersion of S atoms) phasing, a highly redundant dataset at 2.06 Å was collected at 6 different κ-angles with a 360° sweep each, and subsequently initial phases were determined using the autosol.phenix routine[37]. Afterwards, the model was built manually in Coot and with autobuilt.phenix[38,39]. Iterative cycles of refinement between building steps were performed with phenix.refine[40] using the high-resolution dataset (1.7 Å) recorded at a wavelength of 0.97 Å. Restrains of the propargyl moiety were calculated using phenix.elbow[41].

**Ubiquitin-binding assay**. A total of 20 μl Nickel resin (MagneHis™ Protein Purification System, Promega) was saturated with 6His-Smt3 tagged ZUFSP truncations in 200 μl binding buffer (20 mM TRIS pH 7.5, 150 mM NaCl, 20 mM imidazole and 0.1% NP-40) and incubated for 1 h at 4 °C. All truncations containing the catalytic domain were inactivated by a C360A mutation. The resin was washed three times with binding buffer and afterwards incubated with the twofold molar excess of K63-linked ubiquitin chains for 2 h at 4 °C. The washing steps were repeated and the protein was eluted from the beads by addition of 30 μl Laemmli

buffer. The proteins were separated via SDS-PAGE and the subsequent western blots were visualized with α-Smt3 (1:10000; kind gift of J. Dohmen, University of Cologne) or α-ubiquitin P4D1 antibody (1:5000; 3936S; Cell Signaling Technology), respectively.

**Cell culture, transfection, and stable cell line generation**. HEK293T (ATCC® CRL-3216™), HEK293 Flp-In T-REx (ThermoFisher Scientific), and U20s (ATCC® HTB-96™) were maintained by serial passage in Dulbecco's modified Eagle's medium, high glucose (Invitrogen Life Technologies) supplemented with 2 mM of L-glutamine, 1 mM of sodium pyruvate, 1× minimal essential medium non-essential amino acids, 100 U/ml of penicillin, 100 μg/ml of streptomycin (PAA), and 10% fetal calf serum (Biochrom). The stable 3xFLAG-ZUFSP cell line was generated by co-transfection of HEK293 Flp-In T-Rex with pCDNA5 /FRT/TO 3xFLAG-ZUFSP and pOG44 vectors followed by selection with 0.1 mg/ml Hygromycin B (Sigma Aldrich). Single colonies were picked and expression of 3×FLAG-ZUFSP was induced by incubation with 1 μg/ml tetracycline for 24 h. Expression of 3xFLAG-ZUFSP was tested by western blot with α-FLAG M2 antibody (1:3000; F1804; Sigma Aldrich). Transfections for the generation of stable cell lines or transient expression were performed using TransIT-LT1 (Mirus Bio) according to the manufacturer's protocol.

**Immunofluorescence microscopy and laser-induced DNA damage**. For localization studies, U2OS cells on coverslips were fixed with 3% paraformaldehyde 24 h after transfection with the respective constructs. Fixed cells were permeabilized with 0.1% Saponin in 1× phosphate-buffered saline and stained with DAPI (4′,6-diamidine-2′-phenylindole dihydrochloride; Sigma Aldrich) for 15 min. Subsequently, cells were mounted on slides using ProLong Gold antifade reagent and analyzed. Fluorescence images were obtained and processed using a LMS 710 confocal scanning laser microscopy system (Zeiss). For NIR laser irradiation for inducing DNA damage, a Leica TCS SP8 MP-OPO confocal scanning laser microscope was used at the excitation wavelength of 488 nm and a wavelength of 800 nm for inducing the DNA damage. U2OS Cells were seeded in glass-bottomed dishes 35 mm (Ibidi) 2 days before the experiments and transfected with the enhanced green fluorescent protein (eGFP)/mVenus expression vectors 1 day before the DNA damage experiments. Time series of 2 to 3 min were performed by taking a picture every 10 s.

**Purification and MS analysis of interacting proteins**. Flp-In T-REx 293 cells or Flp-in T-REx 293 cells with stable 3xFLAG-ZUFSP integration were seeded in 10 cm dishes and protein expression was induced by addition of 1 μg/ml tetracycline. Alternatively, HEK293T cells were transiently transfected with pCMV2b 3xFLAG-ZUFSP using TransIT-LT1 transfection reagent. At 24 h after induction or transfection, cells were harvested and lysed in lysis buffer (20 mM TRIS pH 7, 150 mM NaCl, 0.1% NP-40, complete protease inhibitor (Roche)) for 30 min at 4 °C, and briefly sonicated before centrifugation at 13,000 × g for 10 min. The supernatant was incubated with prewashed magnetic anti-FLAG M2 beads (Sigma Aldrich) for 2 h at 4 °C. Beads were washed three times with lysis buffer for 5 min. Last washing step was performed with lysis buffer without NP-40. For western blot analysis with rabbit α-FLAG (1:3000; F7425, Sigma Aldrich) and rat α-RPA32 (4E4) (1:1000; 2208S; Cell Signaling Technology) proteins were eluted by addition of 2× Laemmli buffer. For MS analysis, bound proteins were eluted with 200 μg/ml 3xFLAG peptide (Sigma Aldrich) in 6 M urea, 2 M thiourea. Eluted proteins were prepared for mass spectrometry by incubation with 1 mM DTT for 1 h, 55 mM iodoacetamide for 45 min, 0.005 ng/μl Lys-C for 2 h and 0.005 ng/μl Trypsin for 18 h. For LC-MS/MS analysis, an EASY-nLC 1000 chromatograph (Thermo Scientific) was coupled to the quadrupole-based Q Exactive Plus (Thermo Scientific) instrument by a nano-spray ionization source. Peptides were separated on a 50 cm in-house-packed column using a two-solvent buffer system: buffer A (0.1% formic acid) and B (0.1% formic acid in acetonitrile). The amount of buffer B was increased from 7% to 23% within 40 min, followed by an increase to 45% in 5 min, and a washing and re-equilibration step before the next sample injection. The mass spectrometer operated in a Top 10 data-dependent mode, using the following settings: MS1: 70,000 (at 200 *m/z*) resolution, 3e6 AGC target, 20 ms maximum injection time, 300–1750 scan range; MS2: 35,000 (at 200 *m/z*) resolution, 5e5 AGC target, 120 ms maximum injection time, 1.8 Th isolation window, 25 normalized collision energy. Data analysis was performed by MaxQuant software, V 1.5.4.7[42] using the Andromeda search engine[43] against the human proteome reference data (including splice variants) from Uniprot. Default mass tolerance and modification settings were used. Re-quantify, label-free quantification, and match between runs were enabled. 'Oxidation on M', 'Phosphorylation on S,T,Y', and 'GlyGly on K' were allowed as variable modifications. Intensities were averaged over biological triplicates, and the log2 of the intensity ratio 'sample average/control average' was used for enrichment quantification. To account for missing values, pseudocounts corresponding to the minimal observed intensity were added to sample and control averages.

**Electrophoretic mobility shift**. The EMSA assays were carried out as previously described[44] using oligonucleotides described previously to bind SSBP1[28]. In brief, 0.5 μM of ssDNA (5′-GGGCTTCTCCCGCCTTTTTTCCCGGCGGCGGGAGA AGTAGATTGAAG-3′), labeled at the 5′-end with 6-Fam, was incubated with 5

μM of the respective ZUFSP truncation in 40 μl reaction buffer (10 mM TRIS pH 7.5, 0.2 mM DTT, 5 μM ZnSO₄) for 1 h at 4 °C. The DNA-protein complexes were run on an 1.5% agarose gel and visualized by a ChemiDoc MP imaging system (BioRad). The dsDNA was created by annealing the 6-Fam labeled ssDNA with a reverse complementary oligo. In addition, the stem-loop forming oligonucleotides
OriL: 5′ GGGCTTCTCCCGCCTTTTTTCCCGGCGGCGGGAGAAGTAGATTGAAG-3′ and

OriL+6: 5′GGGCTTCTCCCGCCCCCCGCCTTTTTTC CGGCGGCGGGGGCGGGAGAAGTAGATTG-3′
were annealed and used as described above. Gels were stained by SybrGold (NEB) and analyzed on the ChemiDoc system.

**Data availability**. The structure of ZUFSP[232-578] in covalent complex with Ub-PA has been deposited in the Protein Data Bank under the accession code 6EI1. The mass spectrometry data have been deposited to the ProteomeXchange Consortium via the PRIDE partner repository (https://www.ebi.ac.uk/pride) under the accession code PXD008731. All other data supporting the presented findings are available from the corresponding author upon request.

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

## Acknowledgements

This work was supported in part by a DFG grant (SPP1365) to K.H., and by a Vici grant from the Netherlands Foundation for Scientific Research to H.O. We thank Christiane Horst and Claudia Poschner for expert technical assistance and members of the Hofmann lab for discussions. We thank David Komander, Jürgen Dohmen, and Niels Gehring for constructs and reagents. We thank Gerbrand van der Heden van Noort for the synthesis of ubiquitin-AMC. We thank the CECAD proteomics and microscopy facilities, as well as the beamline staff from PETRA III, DESY Hamburg, for support. Funding for the synchrotron visit has been provided by the iNEXT initiative (EU program Horizon 2020).

## Author contributions

T.H. performed most of the biochemical work including crystallization, C.P. helped with crystallization and solved the structure, I.W. contributed to biochemical work and performed functional studies, K.K. contributed to functional studies, K.F.W. and H.O. performed UFM1 work and contributed important reagents, U.B. supervised and helped with the structural work, K.H. conceived the project, performed bioinformatical analyses, and wrote the manuscript. All authors edited and contributed to the manuscript.

## Additional information

**Competing interests:** The authors declare no competing financial interests.

