## [Peer Review File · Nature Communications]

Reviewers' comments:

Reviewer #1 (Remarks to the Author):

This paper identifies a novel deubiquitylase family that shows evolutionary conservation. Humans possess a single family member ZUFSP. It was previously clear this protein was likely a peptidase but its substrate specificity was unclear. In this manuscript the authors convincingly show that ZUFSP and a related protein in *pombe* process ubiquitin chains. Structural elucidation reveals a novel architecture of the catalytic site. After little progress in identifying new DUBs for many years this paper falls a couple of years after a paper which identified a 6th family of DUBs, which has received a lot of attention in the field. The identification of a 7th family is of similar interest. The bioinformatic and structural work is good. The biochemistry is also solid but lacks any quantitative aspect - for a paper describing a novel enzyme- this could be seen as remiss.

Further information is provided on chain specificity, sub-cellular localisation and interaction partners. Some of this is a bit preliminary. For example interaction partners are identified by mass spec only. Whilst the data looks good it is often wise to confirm the findings via an orthologous method and also to look for interactions with endogenous proteins- but maybe the lack of antibody here is prohibitive. I would also worry a bit that the proteins identified are frequently found as contaminants in proteomics experiments (see crapome.org). I would wonder about leaving this out or putting it in supplementary. If the authors keep this in I would like to see the ratio and intensity for the bait protein represented in the data. If normalised intensity is not as high as the prey- this would worry me.

The authors also show recruitment to DNA damage sites, but this is a rather undeveloped observation.

This is a fast moving field right now and I understand the authors may be anxious to publish their findings. I think that publication around the substrate specificity and structural analysis is probably sufficient and fits well with the report style format of Nature Comms.

Reviewer #2 (Remarks to the Author):

The paper by Hermanns and coworkers presents biochemical, structural, and cellular data on two deubiquitinating enzymes which appear to be members of a new family of ubiquitin proteases. These proteins cleave ubiquitin chains but have distinct specificities for ubiquitin chain linkages in vitro. In the human protein, specificity toward K63-linked ubiquitin chains appears to be encoded by a N-terminal set of ubiquitin interacting regions which are lacking in the *S. pombe* member of the family. Cellular interaction studies show potential binding to DNA interacting proteins, which suggest a role for the human enzyme in DNA damage pathways which are regulated by K63-linked chains.

The authors do a good job of clearly connecting their experiments and explaining the data. The comparison between homologs is nicely done and the interesting differences in specificity suggest many interesting pathways that could build on the data presented.

I would like to see the authors address a few questions and concerns in the final paper:

1.) In Figure 2, panels c/d ZUFSP appears to cleave K11 and K48 linked chains, yet the authors do not mention this. The activity is clearly less than the activity on K63-linked chains but some mention of this activity should be noted. Moreover, this specificity is similar to the chain linkage specificity shown for Mug105 which I think brief discussion of this strengthens the connection between these two enzymes. Moreover it supports the role of the N-terminus in specificity.

2.) In Figure 4, the authors correctly note increased activity in the ZUFSP148-578 truncation. Could the increased activity be due to looser specificity toward longer chains? It is not

inconceivable that the N-terminal binding domains allow efficient binding of shorter chains and the apparent increase in activity is simply chain length specificity is now broader. Could the authors look at activity on diubiquitin with the truncation to narrow down the basis for the increased activity?

3.) Related to comment 2, a major point is that the N-terminal region dictate ubiquitin chain linkage specificity, however the specificity for the linkages is never retested with the ZUFSP truncations. The N-terminus could be a scenario similar to the AMSH-STAM complex where these extra ubiquitin binding domains in STAM can dictate linkage and chain length specificity. An activity assay to check this would greatly enhance the arguments the authors are making.

Reviewer #3 (Remarks to the Author):

In this study, the authors identify a seventh family of deubiquitinase that shares a fold with Ufm1 and Atg8 specific proteases, yet cleaves ubiquitin. The authors use an array of biochemical assays to assess the ZUFSP and Mug105's activity against a variety of UbL's and ubiquitin chain types. While the authors provide evidence that these enzymes can cleave Ub-AMC, but not AMC derivatives of SUMO1, SUMO2, NEDD8, or ISG15, neither Ufm1 or Atg8 protease activity is tested. They further show that ZUFSP did not react with Ufm1-PA, while readily reacting with Ub-PA, again consistent with ZUFSP being a DUB. They provide a crystal structure of a portion of one member of this family, Zinc finger with UFM1-specific peptidase domain protein) ZUFSP, in covalent complex with Ub-PA and show that while ZUFSP contains a fold similar to Ufm1 and Atg8 proteases, its active site is more similar to deubiquitinases. While the truncated ZUFSP construct used for crystallization had substantially reduced activity in in vitro DUB assays, the authors contribute this to the absence of N-terminal UBZ-like zinc fingers. Finally, the authors provide evidence that ZUFSP interacts with ssDNA binding proteins and localizes to the sites of DNA damage. This study contains important findings for the ubiquitin community and potentially to the DNA damage field; however, there are some significant concerns that should be addressed.

1) The authors clearly demonstrate that ZUFSP activity against Ub-AMC, but not AMC derivatives of NEDD8, Sumo1, Sumo2, and ISG15, and that ZUFSP reacts with Ub-PA, but not Ufm1-PA. It is not clear if ZUFSP has any activity towards Atg family members, and the authors identify ZUFSP as having significant sequence similarity to Atg4 proteases. Assays to assess Atg protease activity exist (Shu et. al., Autophagy, 2010), and it would strengthen the authors claim that ZUFSP is ubiquitin specific to demonstrate that ZUFSP has no Atg protease activity.

2) The authors mention the remarkable change of Tyr-282/Gln-19 to Ser-351 and note that this alteration of the active site result may limit the catalytic rate of ZUFSP due to it being unable to act as a hydrogen donor. An increase in catalytic rate upon mutation of Ser-351 to Tyr or Gln would provide strong evidence for the importance of this mutation.

3) The authors provide an interesting finding that ZUFSP appears to play a role in the DNA damage response pathway, although it is unclear if this role is significant. Functional assays to support a role for ZUFSP in the DNA damage response would greatly strengthen the manuscript.

Minor points

-There is incorrect referencing of Figure 1 in the text (1b in text is 1c, and the real 1b is never mentioned)

-ZUFSP should be defined when it is first used

Response to the Reviewer Comments

Hermanns et al: A family of unconventional deubiquitinases with modular chain specificity determinants

First, we would like to thank all referees for their careful and constructive criticism of our work. We have addressed all specific comments to the best of our abilities. We hope that will find our revised manuscript sufficiently improved to support it for publication.

The following paragraphs document our responses to the reviewer comments and the resulting changes applied to the manuscript.

Reviewer 1: This paper identifies a novel deubiquitylase family that shows evolutionary conservation. Humans possess a single family member ZUFSP. It was previously clear this protein was likely a peptidase but its substrate specificity was unclear. In this manuscript the authors convincingly show that ZUFSP and a related protein in *pombe* process ubiquitin chains. Structural elucidation reveals a novel architecture of the catalytic site. After little progress in identifying new DUBs for many years this paper falls a couple of years after a paper which identified a 6th family of DUBs, which has received a lot of attention in the field. The identification of a 7th family is of similar interest. The bioinformatic and structural work is good. The biochemistry is also solid but lacks any quantitative aspect - for a paper describing a novel enzyme- this could be seen as remiss.

We have used the fluorometric assay of AMC liberation from RLRGG-AMC, a model substrate cleaved by both ZUFSP and Mug105, to monitor the catalytic reaction and calculate the kinetic properties. As described in the revised 'Result section' on page 4 and documented in the new Supplementary Figure 2b, both reactions follow a Michaelis-Menten kinetics with $K_M=50.4 \mu\text{M}$, $k_{\text{cat}}=4.9 \text{ s}^{-1}$ for ZUFSP, and $K_M=12.22 \mu\text{M}$, $k_{\text{cat}}=7.23 \text{ s}^{-1}$ for Mug105. We also measured the specific activities of ZUFSP and Mug105 in the cleavage of Ubiquitin-AMC. For the human and *S.pombe* DUBs, we found specific activities of 2.3 and 4.1 nmol substrate per mg enzyme per second, respectively. These results are described in the lower part of page 4 and are documented in the new Supplementary Figure 2a.

Further information is provided on chain specificity, sub-cellular localisation and interaction partners. Some of this is a bit preliminary. For example interaction partners are identified by mass spec only. Whilst the data looks good it is often wise to confirm the findings via an orthologous method and also to look for interactions with endogenous proteins- but maybe the lack of antibody here is prohibitive. I would also worry a bit that the proteins identified are frequently found as contaminants in proteomics experiments (see crapome.org).

We too were initially concerned about the validity of our interaction partner findings, given that the RPA proteins are highly abundant nuclear proteins listed in crapome.org as potential contaminants. However, there are three lines of evidence that strongly support our findings. First, while RPA components are consistently found in all our MS data – including the negative controls – they show an 8-16 fold enrichment upon ZUFSP-copurification only in the absence of DNase. This finding already suggests that the RPA proteins not just stick non-specifically to the beads or the vessel walls. More importantly, an independent study (Reference 25 in our manuscript) finds ZUFSP as an interactor of RPA1, RPA2 and RPA3. ZUFSP is a low-abundance protein with a negligible [crapome](http://crapome.org) score and thus very unlikely to be a contaminant. This bi-directional detection provides strong support for this interaction to be real. To further address this issue experimentally, we now also show that ectopically expressed ZUFSP can pull down the *endogenous* RPA complex, as visualized by a commercially available anti-RPA2 antibody. We describe the newly added endogenous pulldown experiment at the bottom of page 8 and document the results in the newly added figure panel 6b.

We have re-phrased the discussion of the published RPA-to-ZUFSP interaction to emphasize the support for our claim. The repeatedly observed DNA-independent interactors USP11 and TCEAL1 are low abundance proteins with low crapome scores and thus unlikely to be contaminants.

I would wonder about leaving this out or putting it in supplementary. If the authors keep this in I would like to see the ratio and intensity for the bait protein represented in the data. If normalised intensity is not as high as the prey- this would worry me.

In our original submission, we had omitted the bait protein from figure 6a since they are 'off scale' and showing them would have made the panels of figure 6a much more bulky without providing much added benefits. However, there is no need to worry: the normalized intensity of the bait is about 5-fold higher than the most abundant prey. In our revised version, we have modified the legend of figure 6a to include the coordinates of the (not shown) bait signal. If required, we can also use a version of figure 6a showing the bait in the upper right hand corner (see below)

The authors also show recruitment to DNA damage sites, but this is a rather undeveloped observation. This is a fast moving field right now and I understand the authors may be anxious to publish their findings. I think that publication around the substrate specificity and structural analysis is probably sufficient and fits well with the report style format of Nature Comms.

The reviewer is correct, we are in a competitive situation with two comparable manuscripts under consideration at Molecular Cell.

Reviewer #2: The paper by Hermanns and coworkers presents biochemical, structural, and cellular data on two deubiquitinating enzymes which appear to be members of a new family of ubiquitin proteases. These proteins cleave ubiquitin chains but have distinct specificities for ubiquitin chain linkages in vitro. In the human protein, specificity toward K63-linked ubiquitin chains appears to be encoded by a N-terminal set of ubiquitin interacting regions which are lacking in the *S. pombe* member of the family. Cellular interaction studies show potential binding to DNA interacting proteins, which suggest a role for the human enzyme in DNA damage pathways which are regulated by K63-linked chains. The authors do a good job of clearly connecting their experiments and explaining the data. The comparison between homologs is nicely done and the interesting differences in specificity suggest many interesting pathways that could build on the data presented.

I would like to see the authors address a few questions and concerns in the final paper:

1.) In Figure 2, panels c/d ZUFSP appears to cleave K11 and K48 linked chains, yet the authors do not mention this. The activity is clearly less than the activity on K63-linked chains but some mention of this activity should be noted. Moreover, this specificity is similar to the chain linkage specificity shown for Mug105 which I think brief discussion of this strengthens the connection between these two enzymes. Moreover it supports the role of the N-terminus in specificity.

We had initially attributed this observation to potential linkage heterogeneity in our synthesized di-Ub chains. Prompted by this reviewer, we have investigated the situation and found our chains to be homogenous. Thus, the observed low-level cleavage of K11 and K48 linkages by ZUFSP and of K11 and K63 linkages by Mug105 is real. In the revised manuscript, we mention the finding in the result section (upper part of page 5 for ZUFSP, middle of page 5 for Mug105). It should be noted that this lack of specificity is only observed for di-Ub cleavage: figure 2e shows hardly any non-K63 cleavage for tetra-ubiquitin chains. Compared to longer chains, even the 'ideal' di-Ub is a very poor ZUFSP substrate in the first place. Thus, we don't think that the residual activity against non-ideal di-Ubs is biologically relevant.

2.) In Figure 4, the authors correctly note increased activity in the ZUFSP148-578 truncation. Could the increased activity be due to looser specificity toward longer chains? It is not inconceivable that the N-terminal binding domains allow efficient binding of shorter chains and the apparent increase in activity is simply chain length specificity is now broader. Could the authors look at activity on diubiquitin with the truncation to narrow down the basis for the increased activity?

As requested by the reviewer, we have tested the ZUFSP¹⁴⁸⁻⁵⁷⁸ (and also the 232-578 truncation) against K63 di-ubiquitin; the result is presented in supplementary figure 3e and shows a marginally increased activity of ZUFSP¹⁴⁸⁻⁵⁷⁸ and a strongly reduced activity of ZUFSP²³²⁻⁵⁷⁸ as compared to full-length protein. Since di-ubiquitin is a very poor ZUFSP substrate, we repeated this experiment on tetra-ubiquitin chains (supplementary figure 3d). Here, too, we see a small activity gain of ZUFSP¹⁴⁸⁻⁵⁷⁸ as compared to the full-length form, best visible at the 60 and 180 min time points. Thus, the slightly increased activity of ZUFSP¹⁴⁸⁻⁵⁷⁸ does not appear to be chain length dependent. In any case, we don't consider this difference to be particularly relevant and have edited the text in the result section (middle of page 7) to better reflect this fact.

3.) Related to comment 2, a major point is that the N-terminal region dictate ubiquitin chain linkage specificity, however the specificity for the linkages is never retested with the ZUFSP truncations. The N-terminus could be a scenario similar to the AMSH-STAM complex where these extra ubiquitin binding domains in STAM can dictate linkage and chain length specificity. An activity assay to check this would greatly enhance the arguments the authors are making.

This is an excellent comment that we should have thought of earlier. We now tested all truncations with higher-than-marginal chain cleavage activity (ZUFSP¹⁴⁸⁻⁵⁷⁸, ZUFSP²³²⁻⁵⁷⁸ and ZUFSP²⁴⁹⁻⁵⁷⁸) against a panel of differently linked tetra-ubiquitin species. Since the latter two truncations only show a weak activity, we had to extend the time course. As shown in the new supplementary figures 3a-c, the truncation leads to a loss of activity, but maintains K63-specificity. Thus, the Zn-finger and MIU regions are required for full activity but not for the K63 preference of ZUFSP. We have included these findings in the result section (middle of page 5).

Reviewer #3: In this study, the authors identify a seventh family of deubiquitinase that shares a fold with Ufm1 and Atg8 specific proteases, yet cleaves ubiquitin. The authors use an array of biochemical assays to assess the ZUFSP and Mug105's activity against a variety of UbL's and ubiquitin chain types. While the authors provide evidence that these enzymes can cleave Ub-AMC, but not AMC derivatives of SUMO1, SUMO2, NEDD8, or ISG15, neither Ufm1 or Atg8 protease activity is tested.

See below.

They further show that ZUFSP did not react with Ufm1-PA, while readily reacting with Ub-PA, again consistent with ZUFSP being a DUB. They provide a crystal structure of a portion of one member of this family, Zinc finger with UFM1-specific peptidase domain protein) ZUFSP, in covalent complex with Ub-PA and show that while ZUFSP contains a fold similar to Ufm1 and Atg8 proteases, its active site is more similar to deubiquitinases. While the truncated ZUFSP construct used for crystallization had substantially reduced activity in in vitro DUB assays, the authors contribute this to the absence of N-terminal UBZ-like zinc fingers. Finally, the authors provide evidence that ZUFSP interacts with ssDNA binding proteins and localizes to the sites of DNA damage. This study contains important findings for the ubiquitin community and potentially to the DNA damage field; however, there are some significant concerns that should be addressed.

1) The authors clearly demonstrate that ZUFSP activity against Ub-AMC, but not AMC derivatives of NEDD8, Sumo1, Sumo2, and ISG15, and that ZUFSP reacts with Ub-PA, but not Ufm1-PA. It is not clear if ZUFSP has any activity towards Atg family members, and the authors identify ZUFSP as having significant sequence similarity to Atg4 proteases. Assays to assess Atg protease activity exist (Shu et. al., Autophagy, 2010), and it would strengthen the authors claim that ZUFSP is ubiquitin specific to demonstrate that ZUFSP has no Atg protease activity.

Prompted by this comment, we have now tested the activity of ZUFSP and Mug105 against LC3A-AMC, a model substrate representing the human Atg8 homolog LC3A. As shown in a new panel of figure 2a, ZUFSP and Mug105 are not active against LC3A-AMC, although LC3A-AMC is readily cleaved by recombinant ATG4B protease (data not shown). This new result is described at the bottom of page 4 in the revised manuscript.

We also generated a propargylated version of LC3B (another human Atg8 homolog). As shown in the new figure 1c, ZUFSP only reacts with Ub-PA but not LC3B-PA, while the Atg8-protease ATG4B reacts with LC3B-PA but not Ub-PA. The description of these results has been added at the top of page 5. Together with the data on UFM1-PA (figure 1b, already present in the original submission) we hope to have convincingly shown that ZUFSP and Mug105 are active against ubiquitin but not the other ubiquitin-like modifiers.

2) The authors mention the remarkable change of Tyr-282/Gln-19 to Ser-351 and note that this alteration of the active site result may limit the catalytic rate of ZUFSP due to it being unable to act as

a hydrogen donor. An increase in catalytic rate upon mutation of Ser-351 to Tyr or Gln would provide strong evidence for the importance of this mutation.

Prompted by this comment, we have now analyzed mutant proteins carrying a Ser351Tyr, Ser351Gln or Ser351Ala mutation. Consistent with our initial assumption that Ser351 is not involved in oxyanion hole formation, the Ser351Ala mutant did not show a decreased activity. However, the mutations of Ser351Tyr or Ser351Gln did not lead to an increase in catalytic rate. This result suggests that the stabilization of the oxyanion hole in ZUFSP is distinct from that of related cysteine proteases. We have included the new data in the new Supplementary figure 5 and changed the main text accordingly (bottom of page 6 and top of page 7 in the revised manuscript).

3) The authors provide an interesting finding that ZUFSP appears to play a role in the DNA damage response pathway, although it is unclear if this role is significant. Functional assays to support a role for ZUFSP in the DNA damage response would greatly strengthen the manuscript.

We fully agree that such data would strengthen our manuscript. However, we must consider that we are in a competitive situation with multiple co-submitted manuscripts overlapping with ours. We know that one manuscript under consideration at Mol. Cell provides evidence for ZUFSP being involved in genome stability, but does not include structural data. The main asset of our manuscript is the structure of the covalent ZUFSP-Ub complex and the mechanistic insights derived from that. Since we are not a DNA repair lab, any detailed study in that direction would cause a substantial delay, which would put us out of competition.

Minor points

-There is incorrect referencing of Figure 1 in the text (1b in text is 1c, and the real 1b is never mentioned)

This oversight has been fixed. Figures 1b and 1c are both referenced correctly now.

-ZUFSP should be defined when it is first used

The definition has been introduced at the first mentioning of ZUFSP in the lower part of page 3.

REVIEWERS' COMMENTS:

Reviewer #1 (Remarks to the Author):

I think this is a decent response from the authors to the collective referees comments and have no further issues with the manuscript.

Reviewer #2 (Remarks to the Author):

The authors did a good job addressing my concerns and in my opinion the concerns of the other reviewers.

Reviewer #3 (Remarks to the Author):

The additional experiments and clarifications have strengthened the manuscript, I have no further concerns.